# An RTK UAV-Based Method for Radial Velocity Validation of Weather Radar

Yubao Chen [1], Lu Li [1,*], Fei Ye [2], Boshi Kang [3], Xiaopeng Wang [1], Zhichao Bu [1], Moyan Zhu [1], Qian Yang [2], Nan Shao [1] and Jianyun Zhang [1]

1    Meteorological Observation Centre, China Meteorological Administration, Beijing 100000, China; chenyb@cma.gov.cn (Y.C.)
2    Changsha Meteorological Radar Calibration Center, Changsha 410205, China
3    Liaoning Meteorological Equipment Support Center, Shenyang 110166, China
*    Correspondence: li-lu@cma.gov.cn

**Abstract:** The quality of weather radar affects the reliability and effectiveness of monitoring severe convective weather. Therefore, rigorous calibration and validation are the foundation for the quantitative application of weather radar. Among the available methods, radial velocity validation is of great significance for reducing the false alarm rate in the identification of tornadoes and thunderstorms. Based on the traditional method that utilizes internal and external instrument radar velocity measurements, we propose a weather radar radial velocity validation method that uses RTK UAV to simulate external targets. In addition, according to the characteristics of the UAV application scenarios, we introduce the evaluation parameter of optimal absolute accuracy to supplement the original parametric system. The experimental results show that the evaluation parameter of optimal absolute accuracy can effectively reduce the interference caused by the systematic deviation of the UAV due to the internal and external environment, which can affect the validation results. When the UAV velocity is not greater than 10 m/s, the optimal absolute accuracy of the radial velocity validation is less than 0.05 m/s, which is essentially consistent with the external instruments' measurement results. This method can be effectively applied to the procedural handling of weather radar radial velocity validation. It is significant for ensuring the accuracy and quality of weather radar radial velocity measurements and improving the effectiveness of radar velocity data applications.

**Keywords:** weather radar; X-band; radial velocity; RTK UAV; validation

## 1. Introduction

Weather radar plays a crucial role in weather monitoring and disaster prevention, providing critical data for predicting and responding to extreme events [1]. In order to deal with severe convective weather that develops rapidly and is potentially disastrous, the China Meteorological Administration (CMA) has constructed a national weather radar network consisting of hundreds of multi-band radar systems. However, the key devices in a radar system may all introduce systematic errors, which increase the uncertainty of the target measurement results [2]. With the continuous development of weather radar, the data application scenarios and demands are moving beyond the traditional qualitative representation and gradually developing toward quantitative monitoring. Radar calibration and validation are the first steps toward improving the quality of the observation data, which is crucial for improving the monitoring and warning capabilities of meteorological targets. Thus, velocity measurement is one of the key parameters of quantization. Due to the complexity of atmospheric conditions and the specificity of the radar system itself, radar radial velocity data may be affected by a variety of errors and disturbances, such as ground clutter, doppler spectrum folding, systematic deviation, etc. In particular, the monitoring of severe convective weather systems, such as tornadoes and thunderstorms, has put forward new requirements for radar velocity measurement capability.

Traditional radial velocity calibration methods for weather radar systems typically use high-precision instruments to measure the internal and external radar system, which can calibrate the receive channel amplitude [3], phase, etc. Existing methods cannot assess the accuracy of moving target velocity measurements under real-world radar operating conditions. Current simulated moving targets mostly use flying platforms mounted with external reference sources, such as tethered balloons, kites, and unmanned aerial vehicles (UAVs) [4]. Compared with other passive carriers, UAVs have powerful flight control systems and real-time kinematic (RTK) systems and can meet the needs of a variety of application scenarios through reasonable route design. Moreover, they are flexible in flight, easy to operate, and can obtain accurate coordinate information in real time [5], which provides a new way of achieving end-to-end calibration and validation for radar systems.

Several researchers have conducted experiments on radar antenna patterns, radar constants, reflectivity factors, differential reflectivity, and radial velocity using "UAV+" and have explored the possibility of using a UAV to calibrate radar systems. Simon et al. constructed a complete antenna pattern using a UAV system, which included the ground clutter, radome, temperature, and other human-introduced external degradation [6]. Arturo et al. constructed a measurement method that utilizes a dual-polarized antenna system based on a UAV and a detection antenna, which achieves the cross-polarization isolation of −40 dB [7–9]. Yin et al. used a UAV combined with the Global Navigation Satellite System, a real-time single-frequency precise point positioning system, to determine the position of the metal ball mounted on the UAV. Through the tracking and scanning of the metal ball by the radar system, the calibration of the weather radar antenna pointing, antenna pattern, and high-precision measurement of the radar constants were realized, and the calculation accuracy of the radar constants was improved [2]. Sun et al. introduced the principle and steps for undertaking the external calibration of a meteorological radar system based on a UAV-mounted metal ball, and the Ku/Ka radar before and after calibration was used to carry out the monitoring of cloud and rain processes and mutual verification of the reflectivity factor [10]. Earle et al. constructed a radar differential reflectivity calibration method based on the UAV with a metal ball and verified that the negative deviations of 6″ and 12″ metal balls on the KOUN WSR-88D radar were −0.56 and −0.52 dB, respectively [11]. Zhu et al. implemented a differential reflectivity calibration of a dual-polarization weather radar system using a UAV and a metal ball, and the results show that the mean Zdr value of a 40 cm metal ball is −0.265 dB [12]. In radar radial velocity measurement, Liu et al. realized the velocity calibration of a vertically pointing millimeter-wave radar system by using a UAV and a metal ball, and determined the absolute deviation between the GPS- and radar-measured results to be a maximum of 0.014 m/s and a minimum of 0.002 m/s [13]. Li et al. carried out calibration of the weather radar reflectivity factor, differential reflectivity, and radial velocity using three types of metal balls suspended from a UAV, and the results showed that the deviation of the measured velocity of a 30 cm metal ball from the GPS-calculated velocity was less than 0.1 m/s at a sampling point number of 128 [14].

Existing methods of weather radar radial velocity external measurement mostly measure the velocity of the metal ball mounted on the UAV in the stationary state. However, the rope suspending the metal ball can be affected by the ambient wind field to produce a pendulum effect, which prevents the metal ball from achieving the ideal stationary state. Moreover, this method can only obtain radar measurements when the target is stationary and cannot simulate a moving target such as a rapidly developing severe convective weather system. Therefore, this paper proposes a weather radar radial velocity validation method based on an RTK UAV, which complements the static calibration of internal and external instrumentation with dynamic validation based on the RTK UAV and realizes systematic weather radial velocity calibration and verification. At the same time, based on the evaluation parametric system of validation for point targets [15], the evaluation parameter of optimal absolute accuracy is proposed. It effectively reduces the validation error caused by the attitude instability of the UAV and improves the robustness of the method and the credibility of the evaluation results. In the following sections, we provide a

detailed introduction to the research methodology, experimental design, and analysis of experimental results, to comprehensively demonstrate our in-depth discussion and scientific research results on weather radar radial velocity validation. Through this series of research work, we expect to provide strong support for improving the quality and credibility of weather radar data and to promote scientific research and technological innovation in the field of meteorology.

## 2. Materials and Methods

The RTK UAV-based weather radar radial velocity validation consists of five key steps (Figure 1). Firstly, the radar environment is checked to confirm the weather conditions and radar configuration parameters. Then, based on the radar position and the far-field region, the geodetic coordinates and altitude of the two endpoints of the UAV routes are theoretically calculated. After completing the preparatory work, the UAV begins to fly along the route at a preset constant velocity, and the radar is set to a tracking mode which performs a synchronized scan of the UAV to check its position and stability. Finally, using time synchronization as a benchmark, the measured radar velocity values are examined in conjunction with the UAV intensity values, and the results of the validation are evaluated both qualitatively and quantitatively and compared with the results of the internal and external instrument calibration.

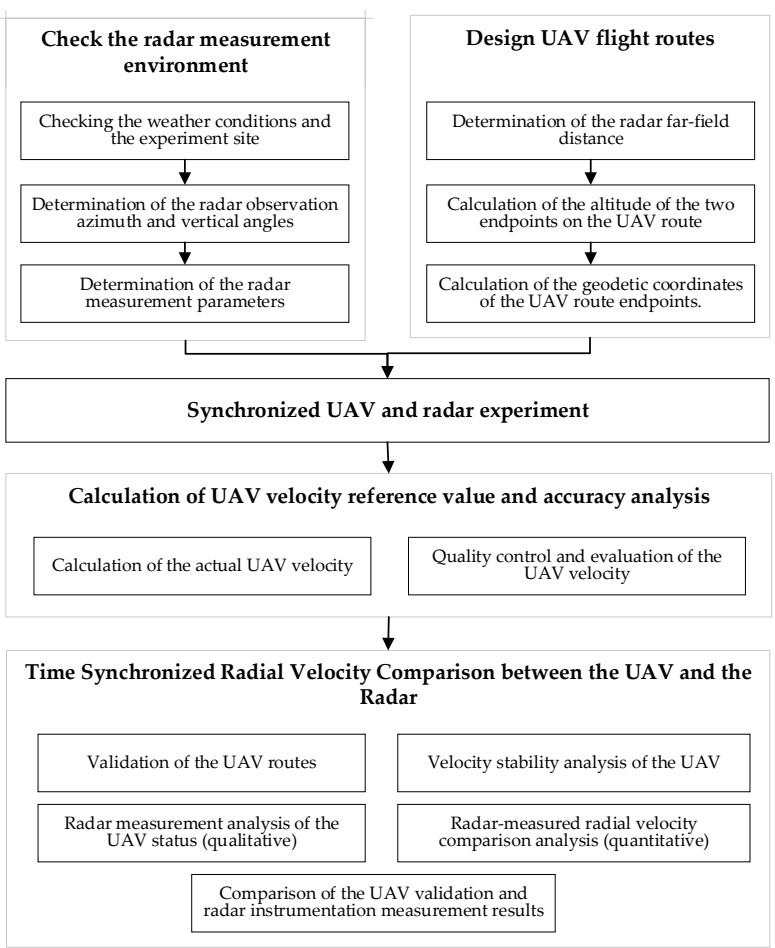

**Figure 1.** Flowchart of the RTK UAV-based radial velocity validation of the weather radar.

### 2.1. Principle of Doppler Radar Velocity Measurement

The Doppler shift, a key parameter in radar measurements, relies on radar frequency and radial velocity. It is directly proportional to the radial velocity and inversely pro-

portional to the radar wavelength. For a Doppler weather radar system operating at a frequency of $f_0$, its wavelength ($\lambda$) is given by

$$\lambda = \frac{c}{f_0} \tag{1}$$

Radar radial velocity, often referred to as Doppler velocity ($V_r$), can be derived from measuring the Doppler shift. This is expressed as

$$V_r = \frac{\lambda f_d}{2} \tag{2}$$

Here, $f_d$ represents the Doppler shift. When a target approaches the radar, the Doppler frequency is positive, indicating that the received signal's frequency is higher than the transmitted one. Conversely, when the target moves away, the Doppler frequency becomes negative.

Additionally, Doppler velocity can be obtained from the rate of change of distance. This calculation is based on the echo phase difference between consecutive pulse pairs, as shown in Equation (3) [16].

$$V_{\max} = \frac{\Delta R}{T} = \frac{\lambda \Delta \varphi}{4\pi T} = \frac{\lambda \Delta \varphi F}{4\pi} \tag{3}$$

where $F$ stands for pulse repetition frequency (PRF), $T$ denotes pulse repetition time (PRT), and $\Delta R$ represents the distance the target moves during $T$.

Considering the discretely sampled nature of radar data, $\Delta \varphi$ can only take values from $-\pi$ to $\pi$. The maximum unambiguous velocity is given by [17]:

$$V_{\max} = \pm \frac{\lambda}{4T} = \pm \frac{\lambda F}{4} \tag{4}$$

The designation "+" indicates movement away from the radar, whereas "−" indicates movement toward the radar.

In pulse-based radar systems, a critical consideration is the pulse repetition time, in order to ensure that all echoes from one pulse return before transmitting the next. This time is crucial to avoid echoes overlapping, which could compromise ranging accuracy. The distance traveled by the electromagnetic wave before the next pulse emission, following the return of all echoes from the previous pulse, is termed the maximum unambiguous distance.

$$R_{\max} = \frac{c}{2F} = \frac{cT}{2} \tag{5}$$

*2.2. UAV Experimental Environment Selection and Radar Parameter Settings*

2.2.1. Checking the Weather Conditions

The UAV-based radial velocity validation of the radar should, as far as possible, be undertaken in clear weather and with a ground wind velocity of level 1–2, reducing the interference of the ambient wind field on the state of the UAV. This could ensure that the measurement results truly reflect the radar performance.

2.2.2. Checking the Experimental Sites

Check the surroundings according to the satellite map, and stay away from electromagnetic interference, mountains, and other dangerous areas. Select a specific area that is appropriate for the operation of the UAV, with a suitable altitude, and apply to the airspace authority with the chosen location.

2.2.3. Determination of the Radar Observation Azimuth and Elevation Angle

Check the radar base data to be measured. Under the conditions permitted by the flight limit, select an area with low background noise and ground clutter interference as the

azimuth and elevation angle for the UAV flight and radar observation, and check that the radar control software has completed the time synchronization of the Beidou timing system.

### 2.2.4. Determination of the Radar Measurement Parameters

Based on the maximum constant velocity of the UAV, parameters such as radar PRF and pulse width are determined. The PRF setting increases with the maximum unambiguous velocity value. Therefore, based on the selected UAV velocity and the maximum unambiguous velocity calculation method of the radar (Equation (4)), the minimum PRF ($F_{min}$) corresponding to the UAV velocity is determined. The PRF of the radar setting needs to be greater than $F_{min}$. This experiment set the maximum velocity of the UAV as 10 m/s, which corresponds to the minimum PRF of about 1253 Hz. Taking into consideration the instability of the UAV velocity when affected by power, environment, etc., in this experiment the PRF was set to 1600 Hz, which corresponds to the maximum unambiguous velocity of 12.8 m/s.

Range resolution is the ability of a radar system to distinguish between two or more targets with the same orientation but at different ranges. Its size depends on the pulse width of the radar system, i.e., a smaller pulse width results in higher resolution. Radar in different bands has varying range resolutions, as shown in Table 1, and thus requires different pulse widths to be set.

**Table 1.** Radar range resolution in different bands.

| Band | S | C | X |
|---|---|---|---|
| Range resolution (m) | 250 | 150 | 75 |

This experiment was conducted using an X-band radar system, and the pulse width required for radar observation was calculated to be 0.5 μs.

$$S_r = \frac{c \cdot \tau}{2} \tag{6}$$

where $S_r$ is the range resolution, $c$ is the velocity of light, and $\tau$ is the pulse width.

### 2.3. UAV Routes Design Based on Radar Geographic Information

### 2.3.1. Determination of the Radar Far-Field Distance

The electromagnetic field is divided into the near field and the far field according to the area of the induction field and radiation field. The correspondence between the electric field and the magnetic field within the near field is more complicated, and the electric field intensity is affected by the change in the target distance. The electric and magnetic field intensity in the far field is inversely proportional to the distance from the center of the antenna. And it has a fixed conversion relationship which is different from the near field [18]. It is commonly assumed that electromagnetic waves in the far field are plane waves, which enables the acquisition of a stable radar cross-section of the target. In fact, electromagnetic waves are actively propagating into space as spherical waves. As the radar system used in this paper has a parabolic antenna, the far-field distance of the antenna is given by the following equation.

$$R = \frac{2D^2}{\lambda} \tag{7}$$

where $R$ is the radar near- and far-field demarcation point (unit: m), and $D$ is the antenna diameter (unit: m). The diameter of the X-band radar antenna used in the experiment is 2.4 m, the frequency is 9.4 GHz, and the wavelength is about 3.2 cm, so the far-field distance of the radar is 360 m.

### 2.3.2. Calculation of the Altitude of the Two Endpoints on the UAV Route

The first step is to determine the slant distances $L_{start}$ and $L_{end}$ between the two endpoints of the route and the radar. The distance of the start point $L_{start}$ needs to be greater than the range of the radar far-field, and $L_{end}$ needs to be determined based on the radar range resolution and the UAV constant motion range bins ($N$ usually $\geq 5$).

$$L_{end} = L_{start} + N \times \text{Resolution} \tag{8}$$

Second, based on the radar elevation angle and the slant distance between the UAV route endpoints and the radar, the actual flight altitude of the two endpoints is calculated. It is worth noting that the experiment required the UAV to always remain within the main radar beam and as close to the center as possible. Therefore, considering the altitude restrictions of the UAV, the theoretically calculated altitude of the main beam center point is determined as the optimal altitude for the UAV route.

As shown in Figure 2, assuming that the radar elevation angle determined by the volume scan data is $\alpha$, the antenna feeds altitude is $H_1$, and $L$ is the slant distance between the UAV route point and the radar, the altitude of the radar beam center point $H_2$ is given by the equation.

$$H_2 = L \sin \alpha + H_1 \tag{9}$$

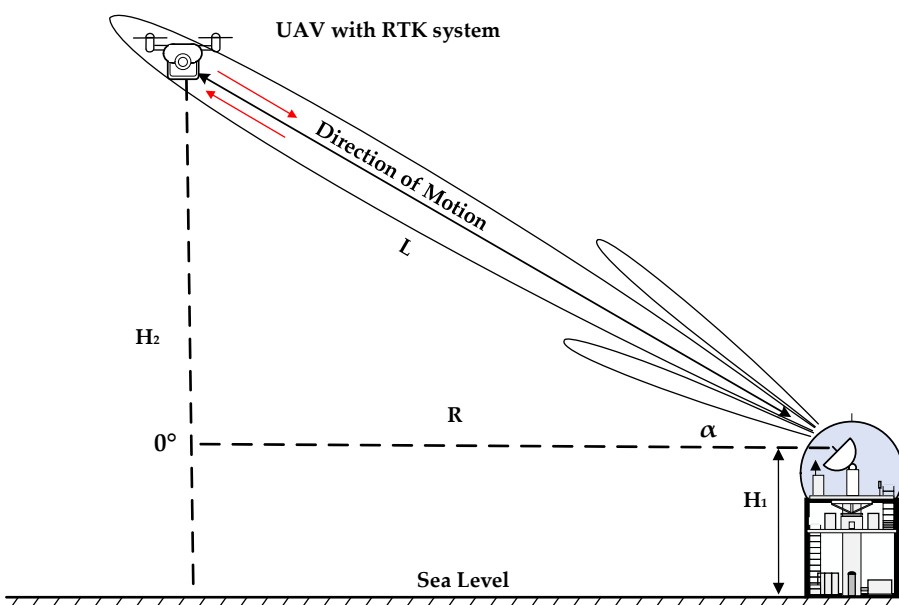

**Figure 2.** Schematic diagram of UAV and radar observation.

It should be noted that the effective input to the UAV is the altitude of the target location, so the UAV route altitude needs to be calculated by incorporating the antenna feed altitude information rather than the actual elevation information.

### 2.3.3. Calculation of the Geodetic Coordinates of the UAV Route Endpoints

If conditions permit, the relative orientation between the UAV route and the radar is preferred to be 0°, 90°, 180°, and 270° to facilitate checking and verification of the route coordinates. With reference to the coordinate calculation method in geodesy, the geodetic coordinates, i.e., latitude and longitude, of the two endpoints of the UAV route at a specific elevation angle are calculated by combining the horizontal distance between the two endpoints of the UAV route and the radar. Knowing the geodesic coordinates of the radar system $(\varphi_1, \lambda_1)$, we can determine that its azimuth angle to the UAV is $\theta$, and the horizontal distance between them is $R = L \times \cos \alpha$. Using the geodesic function of the Python Geopy library, the geodesic coordinates $(\varphi_2, \lambda_2)$ of the route endpoints are obtained

by calculation. Finally, the geodetic coordinates and altitude information are entered into the UAV. The specific calculation codes are shown in the Table 2.

**Table 2.** The calculation of the latitude and longitude of the two endpoints of the UAV route and checking the slant distance between the UAV and the radar.

```
# Create a geodetic coordinate object for the UAV
el = float(input("Please enter radar elevation (°):"))
point1 = Point (radar_lat, radar_lon)

# Calculate the geodesic coordinates of the UAV using the geodesic function
point2 = geodesic (). destination (point1, azimuth, distance_km)

# Calculate UAV altitude
height = distance * math.sin(el * math.pi/180)
UAV_Height = radar_height + height

# Calculate the distance between the radar and the UAV(m)
nodes = [point1, point2]
distance = geodesic (nodes [0], nodes [1]). kilometers*1000
distance = (distance**2+height**2)**(1/2)
print ('Distance between two points:', distance)
```

### 2.4. UAV Velocity Reference Calculation and Accuracy Analysis

The RTK system of the DJI Matrice 300 UAV acquires the real-time position in seconds via GPS, GLONASS, BeiDou, and other satellite navigation systems. The UAV outputs the geodetic coordinates of the target location based on the position and orientation system (POS) data.

#### 2.4.1. Calculation of the Actual UAV Velocity

The POS data include information such as photo name (including time), latitude, longitude, elevation, and three Euler angles (yaw, pitch, roll). Based on the latitude, longitude, and elevation of the two neighboring locations in the POS information, the distance–time method is used to calculate the velocity between the two points.

1. Conversion of latitude and longitude coordinates

As shown in Figure 3, the geodetic coordinate system describes the spatial position of the target using latitude $B$, longitude $L$, and geodetic elevation $H$, whereas the space rectangular coordinate system describes this using $X$, $Y$ and $Z$. Thus, to calculate the distance between two neighboring points, it is necessary to first convert the geodetic coordinate system to the space rectangular coordinate system. For a point $P$ in three-dimensional space, the seven-parameter method in geodesy is used to convert the geodetic coordinate $P = [L, B, H]^T$ into the space rectangular coordinate $P_{xyz} = [X, Y, Z]^T$.

$$\begin{cases} X = (N + H) \cos B \cos L \\ Y = (N + H) \cos B \sin L \\ Z = [N(1 - e^2) + H] \sin B \end{cases} \tag{10}$$

where $N = \frac{a}{\sqrt{1 - e^2 \sin^2 L}}$ represents the local curvature radius of the meridian circle, $a$ denotes the semi-major axis of the Earth ellipsoid, and $e$ signifies the Earth's eccentricity.

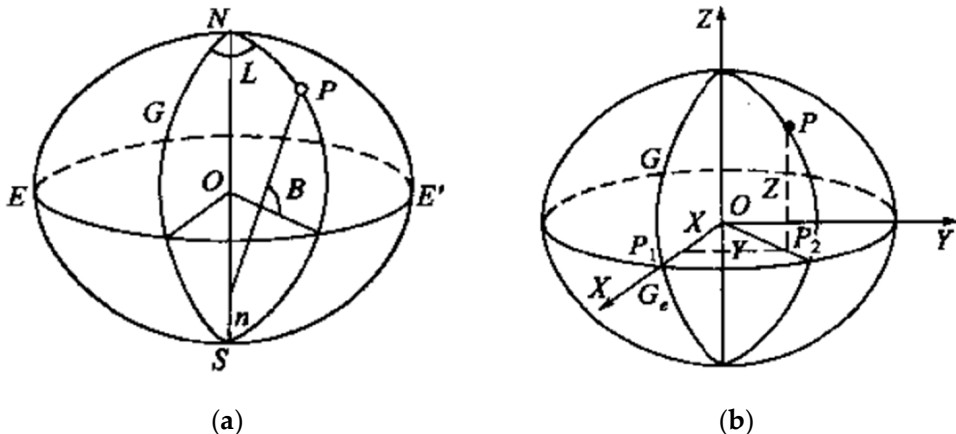

**Figure 3.** Structure of the geodetic coordinate system (**a**) and the space rectangular coordinate system (**b**).

2. Velocity calculation for each position point of the whole UAV route

The geodetic coordinates of two neighboring points of the route obtained from the POS data are transformed to obtain the corresponding space rectangular coordinates $(X_1, Y_1, Z_1)$ and $(X_2, Y_2, Z_2)$. The spatial Euclidean distance between them can be calculated according to the following equation.

$$R = \sqrt{(X_2 - X_1)^2 + (Y_2 - Y_1)^2 + (Z_2 - Z_1)^2} \tag{11}$$

According to the time interval between the two neighboring points, the average velocity is calculated based on the distance–time formula. Since the UAV route is set to fly along the radar radial direction, the results of the following formula are used as a reference value for the radar radial velocity validation.

$$\overline{V_i} = \frac{R}{\Delta T} = \frac{R_n}{T_{n+1} - T_n} \tag{12}$$

where $n$ is taken as 1, 2, 3, etc.

### 2.4.2. Trace Validation of the UAV Actual Route

Using the radar geodetic coordinates and the route information obtained by the RTK, the theoretical elevation of the main beam center corresponding to the UAV at each position point was calculated. By comparing the actual elevation of each position recorded by the UAV, it was verified whether the UAV was flying in the main beam center of the radar system. According to the horizontal distance $R_n$ between the UAV and the radar system, the radar elevation angle $\alpha$ and the antenna feeds elevation $H_R$, the theoretical elevation of each position is $H_n$ and the beam broadening $H_w$ at each point is respectively:

$$H_n = R_n \tan \alpha + H_R \tag{13}$$

$$H_w = R_n \tan(\alpha + \theta/2) - R_n \tan(\alpha - \theta/2) \tag{14}$$

where $\theta$ is the beam width (unit: °). By comparing the actual and theoretical elevations, the height deviation of each point and its percentage of the beam width were calculated.

### 2.4.3. Quality Control of the UAV Velocity

The UAV is set to operate at a constant velocity, but it is difficult to maintain absolute stability due to factors such as wind speed and power influences during flight. In this paper, we calculate the standard deviation of the UAV velocity over the whole route and

analyze the maximum deviation between the actual velocity and the preset velocity to assess whether the operational accuracy meets the standard in the constant velocity state.

*2.5. Time Synchronized Radial Velocity Comparison between the UAV and the Radar*

When the UAV is in a stable state of motion, its radar cross-section tends to remain constant, indicating minimal fluctuations in radar intensity values within a narrow range. When affected by environmental factors such as ambient winds, the three Euler angles of the UAV can be disturbed. It caused changes in the effective radar cross section, which is exposed to electromagnetic waves, resulting in abrupt numerical changes in radar intensity. Therefore, this approach uses the maxima stable dBZ value of the UAV as a reference criterion for selecting radar-validated data. At the same time, the reference values for the UAV velocity are synchronously selected based on the time of the validated data.

The radar system records the velocity values of the UAV during both stationary hover and constant motion along the radar radial direction. A comparison is then made between these recorded velocities and the reference values of the UAV velocity. This paper refers to the evaluation method of validation for point targets and analyzes the results according to two validation criteria: relative accuracy and absolute accuracy. The relative accuracy of the velocity validation results is assessed based on the standard deviation of the radar-measured values of the UAV velocity [15,19,20].

$$\Delta R = \sqrt{\frac{\sum\left(V_{Ri} - \overline{V_R}\right)^2}{N-1}} \tag{15}$$

The equation defines $\Delta R$ as the relative validation accuracy, $V_{Ri}$ as the radar-measured velocity, $\overline{V_R}$ as the average radar-measured velocity, and $N$ as the number of valid data points.

Theoretically, absolute validation accuracy is typically used to explain the absolute differences between measured and reference values. The absolute validation accuracy of the radial velocity is determined by taking the difference between the radar velocity values obtained from the base data and their reference values, using the maximum absolute value of these differences as an indicator [15].

$$\Delta A = Max\left\{\left|V_{Ri} - V_i\right|\right\} \tag{16}$$

Considering that the absolute accuracy indicators are currently used mainly for satellite or airborne SAR, theoretically, the validation results could not be affected by reference targets fixedly deployed on the ground. However, a weather radar is ground-based, and the reference target can only be an airborne target such as a UAV, and as such, it will inevitably be subject to interference caused by the instability of the moving target itself. Moreover, the UAV velocity reference values are derived from the average velocity based on RTK position information, so its proximity to the instantaneous velocity is related to the frequency of positioning updates and the stability of the UAV flight state. To reduce the impact of internal and external environmental factors on the UAV velocity stability, this study, based on valid data pairs, uses the minimum deviation between measured and reference values as the optimal absolute accuracy for validating radar radial velocity.

$$\Delta OA = Min\left\{\left|V_{Ri} - V_i\right|\right\} \tag{17}$$

where $\Delta OA$ is the optimum absolute accuracy, $V_{Ri}$ is the radar-measured velocity values, and $V_i$ is the time-synchronized UAV velocity reference values.

## 3. Experimental Area and Data Source

In July 2023, a radial velocity validation experiment using the X-band weather radar was conducted at the Changsha Meteorological Radar Calibration Center using an RTK UAV. The UAV was used to execute constant velocity flights either toward or away from the radar along the radial direction, while the radar tracked it along its flight route. Synchronized

observational experiments were conducted to obtain radar-measured values of the UAV velocity, and these values were subsequently compared with UAV velocity reference values derived from RTK information. Using the methodology proposed in this paper, the accuracy of X-band weather radar radial velocity measurements can be validated.

### 3.1. Overview of the Experimental Area

The Changsha Meteorological Radar Calibration Center is the only national radar calibration center in China and is based at the Changsha National Climatological Observatory (Figure 4). It is responsible for national research on meteorological radar calibration technology and related operational tasks. Different models of S-, C- and X-band radars have been built around the experimental site. The S-band radar is located about 2.6 km north of the experimental site on Mount Lianhua, and the X-band radar is in the calibration area to the east of the experimental site. In Figure 5, which shows the Digital Elevation Model (DEM) of the experimental site, all the radars are located at elevated positions in mountainous terrain, with relatively little impact from topographical obstacles.

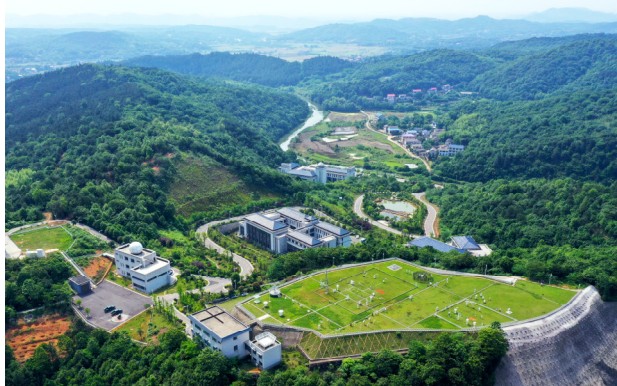

**Figure 4.** Changsha National Climatological Observatory.

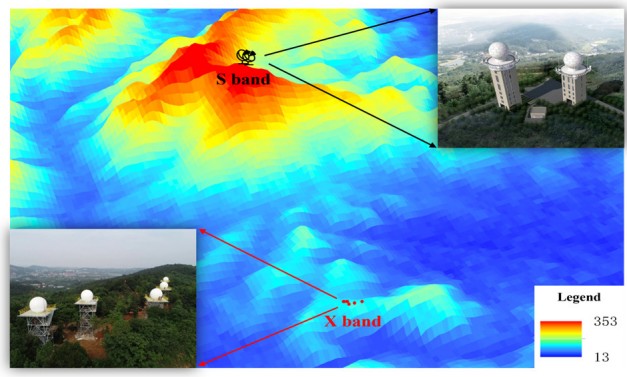

**Figure 5.** S-band and X-band radar calibration site.

### 3.2. The UAV System Technical Parameters

The observation field at the experimental site (Figure 6) was selected as the launching and landing area for the UAV. During the experimental process, the UAV operated in high-precision RTK mode to record position information throughout the whole route. Due to limitations in the UAV's positioning update frequency, data transmission, and storage, the UAV currently stores high-precision position information every 2 s. The main technical parameters of the UAV are shown in Table 3.

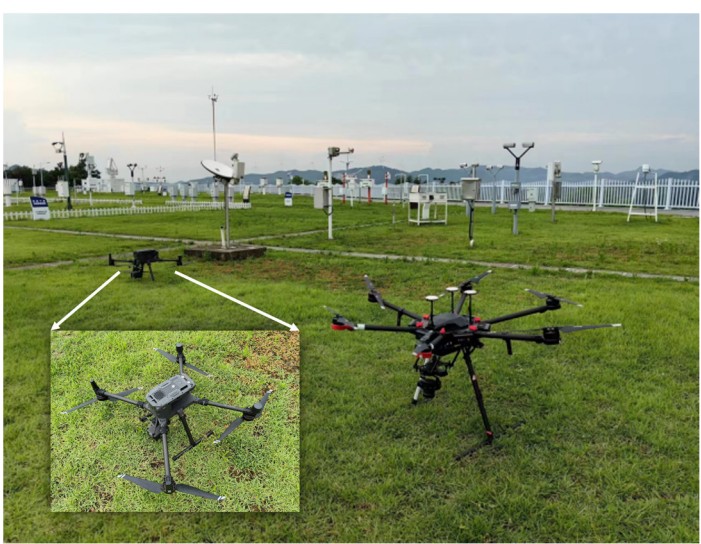

**Figure 6.** The experimental site.

**Table 3.** DJI M300 RTK UAV system main technical parameters.

| Performance Indicators | Parameters |
| --- | --- |
| Frequency | 2.4000–2.4835 GHz 5.725–5.850 GHz |
| Empty weight | 6.3 kg (including dual batteries) |
| Symmetrical motor wheelbase (mm) | 895 |
| Maximum endurance time (min) | 55 |
| Maximum horizontal flight Velocity (m/s) | 17 m/s |
| Maximum altitude (m) | 5000 m |
| Maximum wind resistance (m/s) | 15 m/s (Beaufort scale 7); Maximum permissible velocity during takeoff and landing: 12 m/s |
| RTK positioning accuracy | RTK FIX: 1 cm + 1 ppm (horizontal) 1.5 cm + 1 ppm (vertical) |
| GNSS | GPS, GLONASS, BeiDou, Galileo |

### 3.3. Radar Control Parameter Setting

The X-band radar was set to track scanning mode with a pulse width of 0.5 μs, a range resolution of 75 m, a sample size of 128, a single PRF, and the deactivation of clutter filters such as ground clutter. Due to the proximity of the UAV experimental site to the radar, the effects of ground clutter interference were significant in the absence of filters. Therefore, the elevation angle of the radar was set by referencing the volume scan data where the background clutter was relatively low. The specific observation parameters are shown in Table 4.

**Table 4.** X-band radar system main configuration parameters.

| Indicators | Parameters | Indicators | Parameters |
| --- | --- | --- | --- |
| Wavelength (cm) | 9.4 GHz | Number of pulses | 128 |
| Scanning mode | TRACK | Antenna beamwidth (°) | 1 |
| Pulse width (μs) | 0.5 | Elevation angle (°) | 8.5 |
| Maximum unambiguous velocity (m/s) | 12.8 | Ground clutter suppression filter | OFF |

## 4. Results

### 4.1. Design and Validation of UAV Routes

The experiment obtained volume scan data within a 3 km range under clear sky conditions, with the aim of finding a flight area with a relatively clean background and minimal interference from ground clutter. To ensure the safety of the UAV operation, the experiment avoided Mount Lianhua in the north of the X-band radar calibration site. Instead, 180° was used as the experimental azimuth. After selecting the experimental orientation, the information from each elevation angle layer of the volume scan data was analyzed. The red box in Figure 7 shows that there was a target with an intensity of more than 10 dB at an elevation angle of 6.0° in the selected orientation, whereas at 9.9° the intensity of the same target was less than 5 dB. In view of the maximum flight altitude of the UAV being 500 m, an actual radar observation elevation angle of 8.5° was ultimately selected for the experiment.

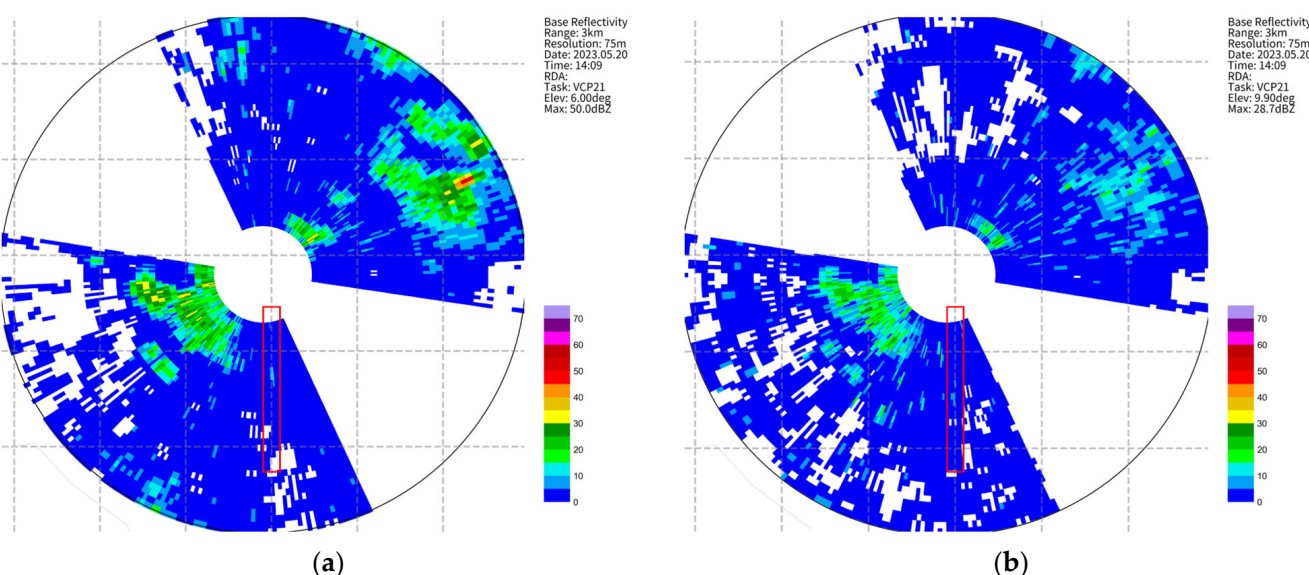

**Figure 7.** X-band radar volume scan data. (**a**) Elevation angle: 6.0°; (**b**) Elevation angle: 9.9°.

The flight route of the UAV was required to meet the far-field condition of the X-band radar at 360 m and maintain a constant velocity over an elevation angle of 8.5°, using the antenna feeds as a reference. Therefore, the start point for the route was chosen at a slant range of 650 m from the radar. The UAV flew at a constant velocity over 600 m, covering eight range bins, and finished at a slant range of 1250 m from the radar. With the DEM data of the experimental site taken into consideration (Figure 8), the flight route effectively avoided the surrounding mountainous terrain, ensuring the safety of the round-trip flight routes.

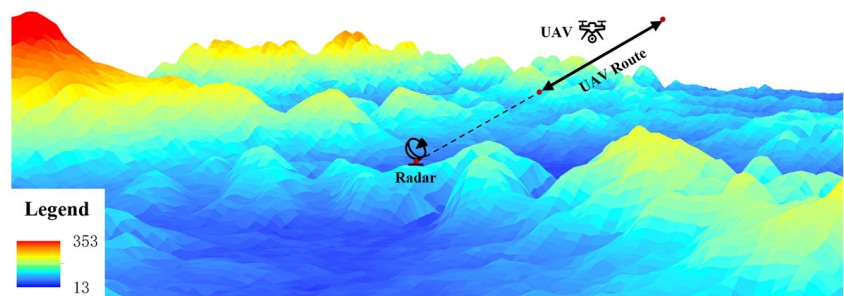

**Figure 8.** Radial velocity validation of the UAV routes.

To verify the relative position between the UAV route and the radar, the experiment compared the actual flight elevation of the UAV with the elevation of the radar main beam center. Based on the latitude, longitude, and radar elevation angles collected by the RTK system on the UAV, and with reference to the geometric position relationships shown in Figure 2, the experiment calculated the elevation of the radar main beam center and the beam widths for each point. We define the deviation between the UAV's actual flight elevation and the elevation of the radar main beam center as the elevation deviation. Combining Table 5 and Figure 9 for different velocities, the maximum ratio of the elevation deviation to the beam width occurs near the proximal end of the route, which is closer to the radar system. The maximum elevation deviation is 1.349 m, which is 11.47% of the beamwidth at this point. As shown in Table 6, the minimum ratio is at the far end of the route, with a minimum elevation deviation of 1.167 m, which is 5.72% of the beamwidth at this point. This indicates that the difference between the actual flight elevation of the UAV and the elevation of the radar main beam center is relatively stable. As the distance from the radar increases, the beam width increases, and the effect of the UAV elevation deviation on the radar observations decreases. As shown in Figure 9, the UAV remains within the radar main beam throughout the flight. Furthermore, the route is generally parallel to the main beam center. Therefore, the actual route is essentially completed along the radial direction of the radar.

**Table 5.** The maximum deviation between the UAV elevation and the radar main beam center.

| Velocity (m/s) | Slant Range between UAV and Radar (m) | Elevation of Main Beam Center (m) | UAV Flight Elevation (m) | Beamwidth (m) | Maximum Deviation (m) | Percentage of Beamwidth Occupied (%) |
|---|---|---|---|---|---|---|
| +7 | 672.765 | 216.944 | 215.595 | 11.764 | 1.349 | 11.47 |
| −10 | 673.036 | 216.985 | 215.723 | 11.769 | 1.262 | 10.72 |
| +8 | 703.172 | 221.489 | 220.153 | 12.296 | 1.336 | 10.86 |
| −10 | 663.880 | 215.617 | 214.347 | 11.609 | 1.270 | 10.94 |

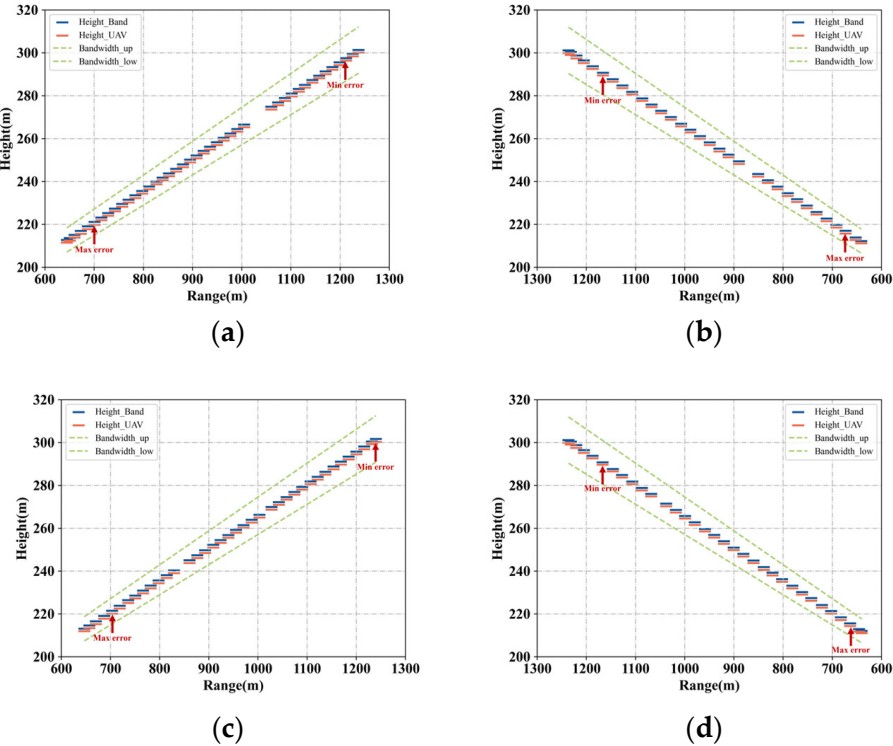

**Figure 9.** Elevation difference between UAV flight route and radar main beam center. (**a**) 7 m/s; (**b**) 10 m/s; (**c**) 8 m/s; (**d**) 10 m/s.

**Table 6.** The minimum deviation between the UAV elevation and the radar main beam center.

| Velocity (m/s) | Slant Range between UAV and Radar (m) | Elevation of Main Beam Center (m) | UAV Flight Elevation (m) | Beamwidth (m) | Maximum Deviation (m) | Percentage of Beamwidth Occupied (%) |
|---|---|---|---|---|---|---|
| +7 | 1211.933 | 297.524 | 296.337 | 21.193 | 1.187 | 5.60 |
| −10 | 1166.227 | 290.693 | 289.511 | 20.393 | 1.182 | 5.79 |
| +8 | 1231.907 | 300.509 | 299.265 | 21.542 | 1.244 | 5.77 |
| −10 | 1167.008 | 290.809 | 289.643 | 20.407 | 1.167 | 5.72 |

### 4.2. Analysis of UAV Velocity Stability

When a UAV operates in airspace, it experiences vibrations due to various factors such as ambient wind and aircraft dynamics. Even if the UAV is set to operate at a constant velocity, there may still be differences between the actual motion velocity and the preset velocity. To ensure the credibility of velocity reference values, the experiment verified the stability of the UAV state at different preset velocities in approximately the same environment. From Figure 10, it is evident that when the UAV operates at a velocity greater than 5 m/s, the velocity standard deviation is consistently greater than 1 m/s, and the deviation between the actual average velocity and the preset velocity is within 1 m/s. Throughout the whole route, the overall velocity variation shows a pattern of acceleration-steady-deceleration. Excluding non-steady states during the acceleration and deceleration of the UAV, the deviation between the actual average velocity and the preset velocity is less than 0.1 m/s. The standard deviation increases with the higher preset velocity of the UAV. When the preset velocity is 10 m/s, the velocity standard deviation is approximately 0.5 m/s. By eliminating velocity reference values during non-steady states, it is possible to significantly reduce the system bias that unavoidably exists in the UAV due to internal and external environmental factors. It provides a reliable data guarantee for conducting radar validation on UAVs in motion.

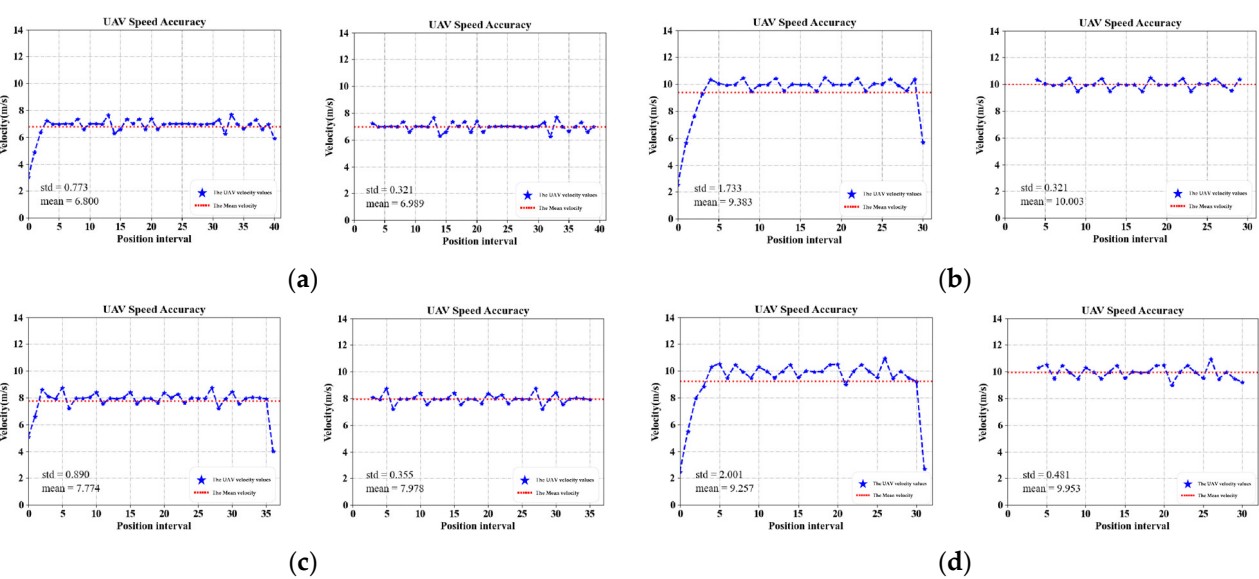

**Figure 10.** Analysis of the UAV velocity excluding the acceleration and deceleration process. (**a**) 7 m/s; (**b**) 10 m/s; (**c**) 8 m/s; (**d**) 10 m/s.

### 4.3. Analysis of Radar Measurements of UAV States

The UAV flew continuously along the route and the radar performed track scanning of the specified target. Using the continuous radar base data corresponding to routes 3 and 4 as examples, we analyzed the actual operational state of the UAV. As can be seen from Figure 11, the UAV first hovered and then flew at a constant velocity from the start point

at 650 m to the endpoint at 1250 m. After reaching the endpoint, the UAV hovered for a while and then returned along the same route at different velocities. Throughout the entire operation, the intensity of the UAV was consistently above 45 dB, and its return velocity was faster than its forward velocity. This indicates that the radar measurement results are consistent with the actual motion state of the UAV and that the radar antenna remained focused on the UAV.

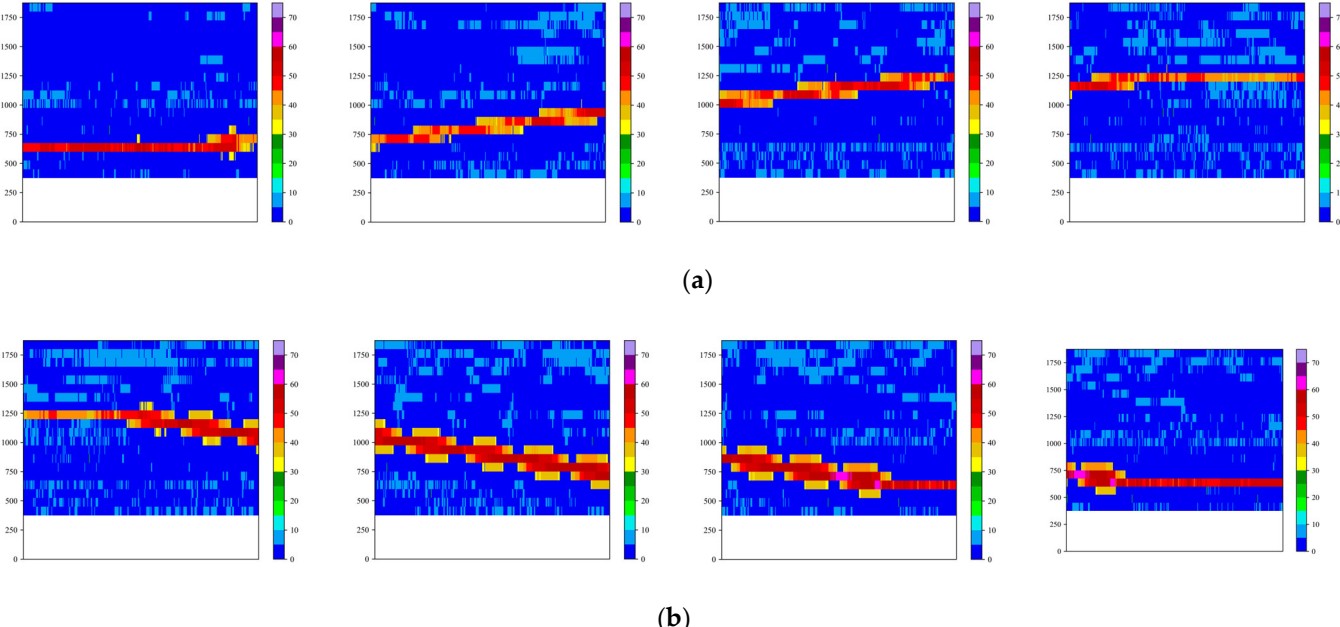

**Figure 11.** Radar measurement results of the UAV flight status on round-trip routes. (**a**) 8 m/s; (**b**) 10 m/s.

Qualitative analysis of the radar velocity measurement results shows that the UAV repeatedly underwent a state transition from hovering to low velocity away, then to a high-velocity approach. For better visual interpretation, the experiment emphasized the display of effective data within a range of ±0.5 m/s for each preset velocity. Based on Figure 12, pixels with UAV intensity values greater than 45 dB are identified as target pixels. Whether the UAV is hovering, moving away from the radar, or approaching the radar, the target pixels are generally close to the preset velocity. Furthermore, the closer the actual measurement results are to the preset velocity, the denser the distribution of target pixels. Therefore, the radar measurement results effectively reflect the velocity state of the UAV, indicating that the trend of the actual radar-measured velocity is generally consistent with the preset velocity of the UAV.

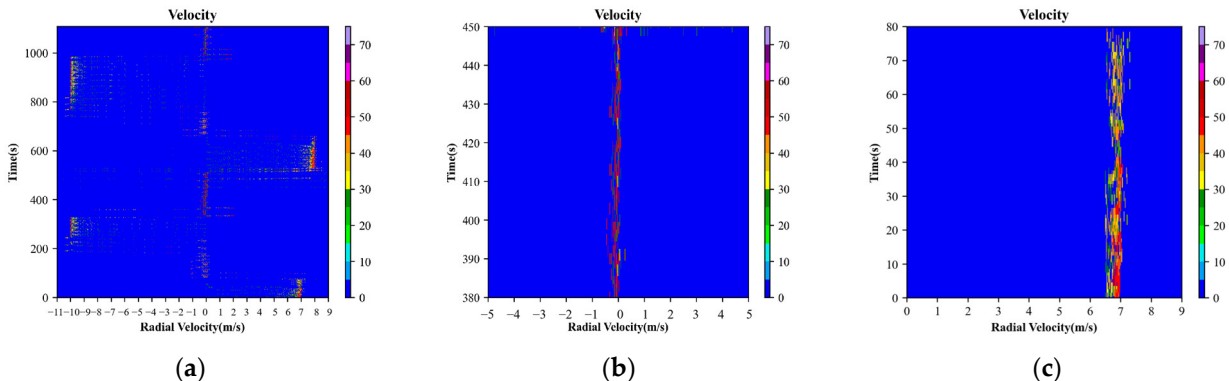

**Figure 12.** *Cont.*

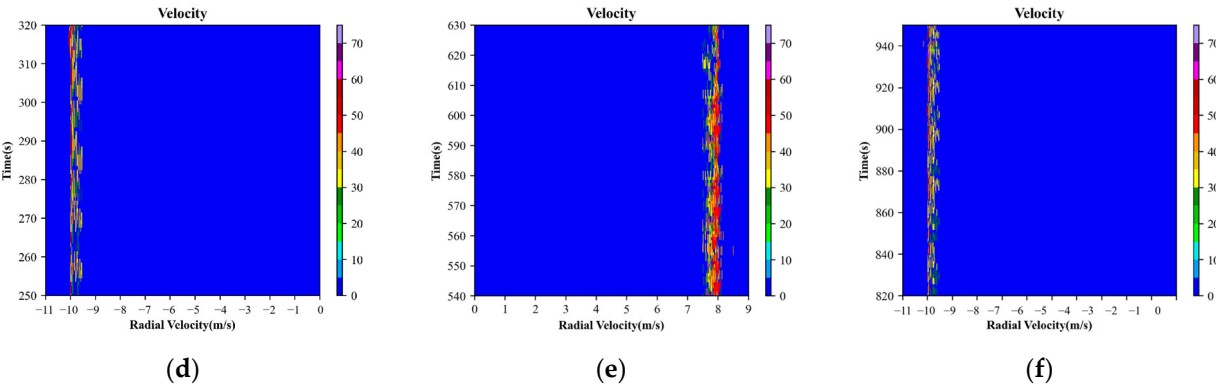

(**d**) (**e**) (**f**)

**Figure 12.** Radar monitoring and analysis of the velocity state of the UAV. (**a**) The whole process; (**b**) hover; (**c**) 7 m/s; (**d**) −10 m/s; (**e**) 8 m/s; (**f**) −10 m/s.

### 4.4. Comparative Analysis of Radar-Measured Radial Velocity

When the UAV is hovering, the radar-measured velocity should theoretically be 0 m/s. However, the attitude of the UAV is influenced by factors such as wind speed and the rotor, which could alter velocity components along the radar radial direction, affecting the measurement results. Therefore, the experiment first compared the velocity accuracy in the hovering state of the UAV and then analyzed the measurement instability introduced by the attitude of the UAV. As shown in Figure 13, when the UAV velocity varies by less than 0.2 m/s, the corresponding intensity values also show small variations, and the target RCS remains relatively constant. However, if the UAV velocity varies significantly (as highlighted by the red box in the figure below), there is a difference of approximately 10 dB in the intensity of the target. The standard deviation of the measured radial velocity is 0.135, with a mean value of −0.069 m/s, according to the complete recorded data in the hovering state. Therefore, the more stable the UAV state, the higher the quality of the corresponding velocity measurement. Conversely, if the UAV RCS varies significantly, the influence of the UAV attitude on the radar measurement results will be more evident. This further validates the correlation between target velocity and intensity in the radar signal processing algorithm.

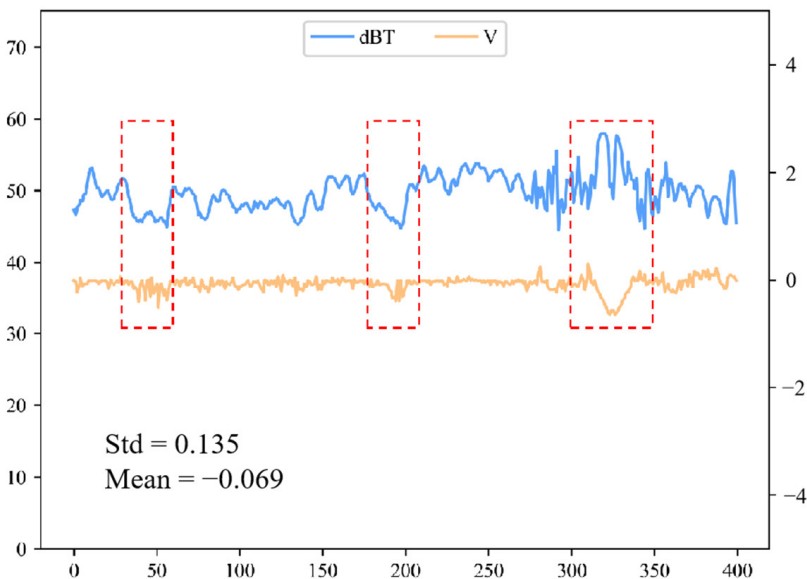

**Figure 13.** The UAV-measured velocity while hovering.

According to the UAV intensity and velocity rule observed during the hovering state, we developed a valid data-filtering method based on non-zero velocity using the target intensity. First, three contiguous range bins were selected for each route. Then, within each range bin, five pairs of data were selected with continuous stable values near the maxima that did not vary by more than 1 dB. These selected data were considered the effective measured data for radial velocity comparison. At the same time, taking into consideration the random jitter in the UAV velocity, data periods with relatively stable UAV velocities were selected during the comparison to ensure the reliability of the velocity reference values. Based on stable and effective comparison data pairs, the minimum absolute deviation between the UAV velocity reference values and the radar-measured values was used as the optimal absolute accuracy for the radial velocity validation. By calculating the standard deviation of the actual measured velocities on the same route, the relative accuracy of the weather radar velocity validation was assessed. The scientific validity of the validation method proposed in this study and the rationality of optimizing the evaluation indicators were analyzed using radar-measured data.

We analyze the data in detail in Table 7. Analyzed from a quantitative point of view, the maximum deviation at +7 m/s was 0.04 m/s and the minimum deviation was 0.02 m/s; the maximum deviation at +8 m/s was 0.22 m/s and the minimum deviation was 0.04 m/s. This is because when the UAV is working at lower velocities, the reference velocity variations are minimal, resulting in a larger continuous and stable data period. This increases the reliability of the reference values during synchronous comparisons. Conversely, at a velocity of $-10$ m/s, the minimum deviation was no greater than 0.05 m/s, and the maximum deviation approached 0.5 m/s. This is attributed to the inadequate stability of the UAV internal state at high velocities, resulting in abrupt changes in adjacent velocity reference values of approximately $\pm0.5$ m/s. Moreover, the maximum deviation could potentially include a systemic bias associated with velocity discontinuity. Therefore, this study optimizes traditional validation assessment methods by using the minimum deviation between the measured and reference values of the UAV velocity as the evaluation criterion for optimal absolute accuracy. It can effectively reduce the inspection error introduced by the system deviation of the UAV. At the same time, even when the UAV was in a relatively unstable high-velocity state, the relative accuracy of the radar radial velocity validation remained less than 0.2. Furthermore, the optimum absolute accuracy of synchronous comparisons between the UAV and radar was no greater than 0.05 m/s. This conclusively demonstrates that weather radar radial velocity validation based on the RTK UAV can quantitatively reflect the radar velocity measurement capabilities and that the method has good robustness.

Based on Figure 14b,d, it is evident that as the velocity increases, the UAV intensity and velocity values measured by the radar become more stable and smoother. Conversely, at lower velocities (Figure 14a,c), the fluctuations in these values become more pronounced. Based on the simulation of the target intensity using the sinc function, as shown in Figure 15, it can be observed that a slower velocity corresponds to a longer duration at the same distance. There is also an increase in the number of sampled points involved in the weighted average. The amplitude variation of the sampled points becomes relatively larger, leading to more pronounced fluctuations in the processed actual intensity values. Conversely, if the target velocity is higher, resulting in a shorter duration, the number of sampled points involved in the weighted average is reduced. As a result, the processed actual intensity values have relatively smoother characteristics. This empirical pattern is consistent with the sampling principles inherent in radar signal processing for target analysis.

**Table 7.** Radar radial velocity comparison results based on the UAV. (Unit: m/s).

| Preset V¹ | Range (m) | Maxima dBT | Radar V | UAV V² | Difference | Preset V¹ | Range (m) | Maxima dBT | Radar V | UAV V² | Difference |
|---|---|---|---|---|---|---|---|---|---|---|---|
| +7 | 825 | 47.11 | 6.98 | 7.00 | 0.02 | −10 | 975 | 57.59 | −9.95 | 9.80 | 0.15 |
| | | 47.47 | 6.94 | 7.00 | 0.06 | | | 57.86 | −9.95 | 9.80 | 0.15 |
| | | 47.97 | 6.98 | 7.00 | 0.02 | | | 57.95 | −9.95 | 9.80 | 0.15 |
| | | 47.75 | 6.92 | 7.00 | 0.08 | | | 57.84 | −9.95 | 9.80 | 0.15 |
| | | 47.79 | 6.98 | 7.00 | 0.02 | | | 58.15 | −9.95 | 9.80 | 0.15 |
| | 900 | 47.38 | 7.03 | 7.00 | 0.03 | | 900 | 58.59 | −9.95 | 9.97 | 0.02 |
| | | 47.47 | 6.9 | 7.00 | 0.10 | | | 58.84 | −9.98 | 9.97 | 0.01 |
| | | 47.54 | 6.96 | 7.00 | 0.04 | | | 58.95 | −9.95 | 9.97 | 0.02 |
| | | 47.86 | 6.96 | 7.00 | 0.04 | | | 58.9 | −9.95 | 9.97 | 0.02 |
| | | 47.11 | 6.92 | 7.00 | 0.08 | | | 59.02 | −9.95 | 9.97 | 0.02 |
| | 975 | 51.86 | 6.96 | 7.00 | 0.04 | | 825 | 59.2 | −9.98 | 9.50 | 0.48 |
| | | 52.13 | 6.94 | 7.00 | 0.06 | | | 59.47 | −9.98 | 9.50 | 0.48 |
| | | 51.83 | 6.96 | 7.00 | 0.04 | | | 59.33 | −9.95 | 9.50 | 0.45 |
| | | 51.29 | 6.96 | 7.00 | 0.04 | | | 59.27 | −9.94 | 10.05 | 0.11 |
| | | 51.47 | 7 | 7.00 | 0.00 | | | 59.09 | −9.94 | 10.05 | 0.11 |
| Deviation | Maximum / Minimum | | 0.04 / 0.02 | Relative Accuracy | 0.03 | Deviation | Maximum / Minimum | | 0.48 / 0.02 | Relative Accuracy | 0.17 |
| Preset V¹ | Range (m) | Maxima dBT | Radar V | UAV V² | Difference | Preset V¹ | Range (m) | Maxima dBT | Radar V | UAV V² | Difference |
| +8 | 825 | 48.08 | 7.96 | 8.22 | 0.26 | −10 | 1200 | 55.5 | −10.07 | 10.34 | 0.27 |
| | | 48.43 | 8.05 | 8.22 | 0.17 | | | 55.93 | −10.04 | 10.34 | 0.30 |
| | | 48.15 | 7.9 | 8.22 | 0.32 | | | 55.9 | −10.04 | 10.34 | 0.30 |
| | | 48.77 | 8 | 8.22 | 0.22 | | | 55.83 | −10.03 | 10.45 | 0.42 |
| | | 48.18 | 7.96 | 8.22 | 0.26 | | | 56.06 | −10.03 | 10.45 | 0.42 |
| | 900 | 54.54 | 7.9 | 8.01 | 0.11 | | 1125 | 56.27 | −9.94 | 10.21 | 0.27 |
| | | 54.81 | 7.88 | 8.01 | 0.13 | | | 56.15 | −9.94 | 10.21 | 0.27 |
| | | 54.93 | 7.9 | 8.01 | 0.11 | | | 56.59 | −9.91 | 10.21 | 0.30 |
| | | 54.75 | 7.9 | 8.01 | 0.11 | | | 56.52 | −9.91 | 10.21 | 0.30 |
| | | 54.24 | 7.92 | 8.01 | 0.09 | | | 56.68 | −9.9 | 10.21 | 0.31 |
| | 975 | 53.95 | 7.96 | 7.96 | 0.00 | | 1050 | 57.27 | −9.95 | 9.99 | 0.04 |
| | | 53.93 | 7.96 | 7.96 | 0.00 | | | 57.47 | −9.94 | 9.99 | 0.05 |
| | | 54.04 | 7.98 | 7.96 | 0.02 | | | 57.43 | −9.95 | 9.99 | 0.04 |
| | | 54.08 | 7.98 | 7.96 | 0.02 | | | 57.47 | −9.94 | 9.99 | 0.05 |
| | | 54.34 | 8 | 7.96 | 0.04 | | | 57.56 | −9.94 | 9.99 | 0.05 |
| Deviation | Maximum / Minimum | | 0.22 / 0.04 | Relative Accuracy | 0.18 | Deviation | Maximum / Minimum | | 0.42 / 0.05 | Relative Accuracy | 0.14 |

V¹ represents the preset velocity of the UAV; V² represents the actual velocity value of the UAV.

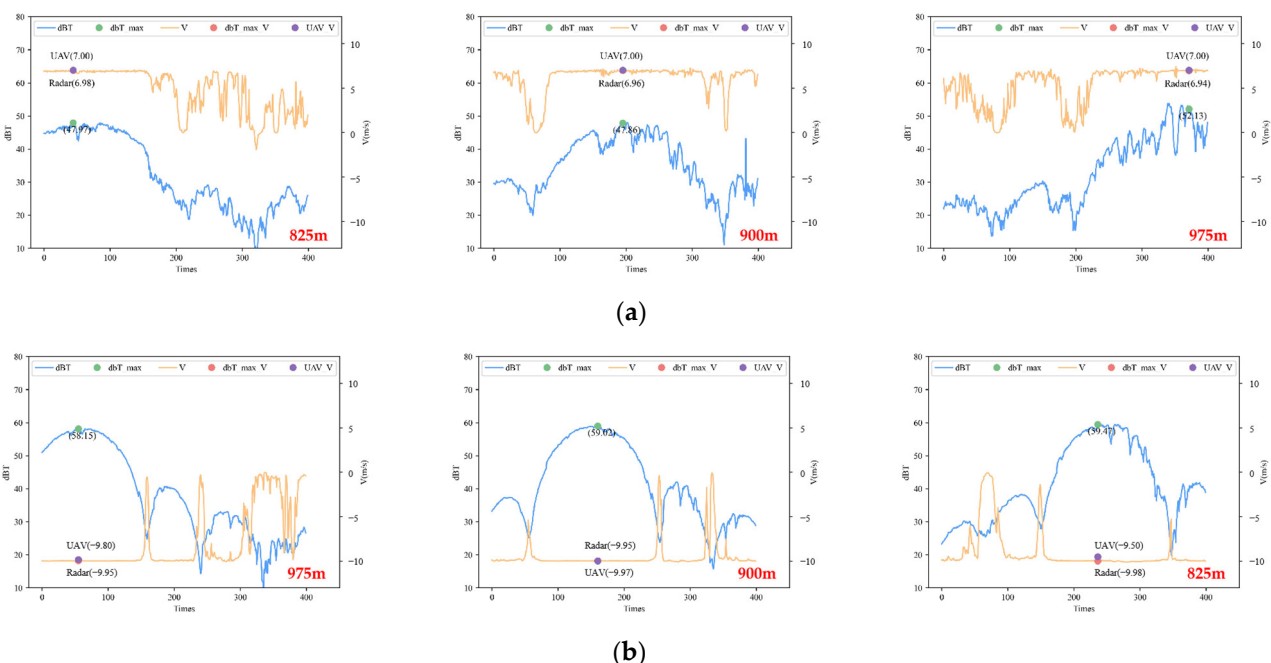

**Figure 14.** *Cont.*

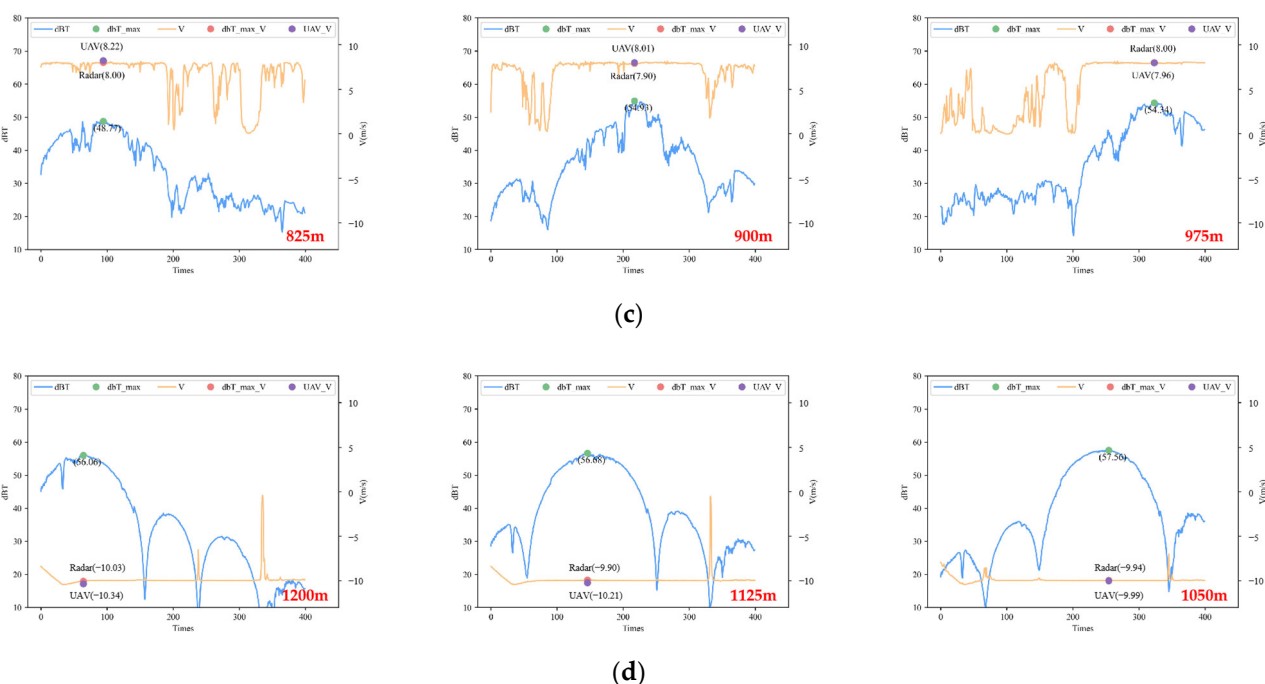

**Figure 14.** The weather radar radial velocity comparison results. (**a**) 7 m/s; (**b**) 10 m/s; (**c**) 8 m/s; (**d**) 10 m/s.

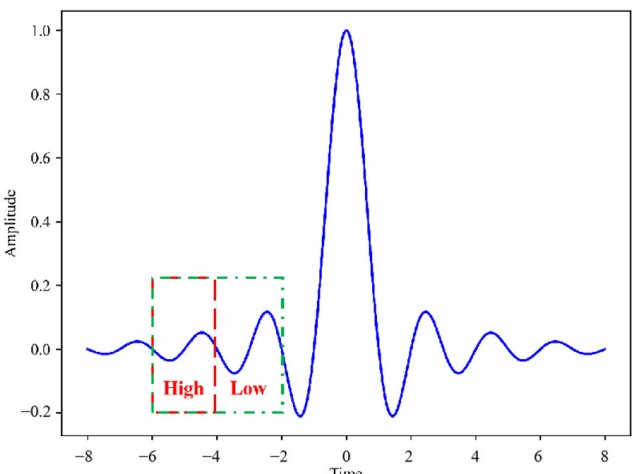

**Figure 15.** Target intensity simulation based on the sinc function. The red box represents the sample time and amplitude fluctuations in the high-velocity case, and the green box corresponds to the sample time and amplitude fluctuations in the low-velocity case.

### 4.5. Comparison of the UAV Validation and Radar Instrument Measurement Results

In the actual application of operational radars, the performance index and observation effect of the S-band radar were better than those of the X-band radar system. To objectively evaluate the performance of the X-band radar system and improve its observation effect, this paper refers to the meteorological industry standard of the People's Republic of China, "S-band Dual-Line Polarimetric Doppler Weather Radar", which stipulates that the error in radial velocity measurement should not be more than 1 m/s. In addition, the experiment uses the measured radial velocity results from both internal and external instruments as a reference for validation. As shown in Table 8, under single-frequency conditions, the maximum deviation of the external velocity measurement is 0.04 m/s, whereas the maximum deviation of the internal measurements is 0.1 m/s. The optimal absolute accuracy

of the radar radial velocity validation based on the RTK UAV is in close agreement with external instrument measurements, as shown in Table 9. In summary, both the traditional relative and absolute accuracy (the maximum deviation) and the optimal absolute accuracy index can meet the requirements of ±1.0 m/s radial velocity measurement, which further verifies the feasibility and effectiveness of the proposed method.

**Table 8.** Measurements of radial velocities of instruments in and out of the radar (unit: m/s).

| Validation Content | | | Indicator | Result | Comment |
|---|---|---|---|---|---|
| Radial velocity measurement validation | Maximum deviation in a single frequency velocity measurement | Horizontal Vertical | ±1.0 | 0.04 −0.04 | External |
| | Maximum deviation in velocity measurement | | ±1.0 | 0.1 | Internal |

**Table 9.** Comparison of the UAV validation and instrumentation measurements (unit: m/s).

| Velocity \ Indicator | The Maximum Deviation | Difference | The Minimum Deviation | Difference |
|---|---|---|---|---|
| +7 | 0.04 | 0 | 0.02 | −0.02 |
| −10 | 0.48 | 0.44 | 0.02 | −0.02 |
| +8 | 0.22 | 0.18 | 0.04 | 0 |
| −10 | 0.42 | 0.38 | 0.05 | 0.01 |

## 5. Conclusions

The accurate measurement of weather radar radial velocity is of paramount importance for quality control and applications of radar velocity data. Traditional methods for measuring radial velocity in weather radar often involve the use of high-precision instruments for internal and external static calibration. These calibrations focus on adjusting parameters such as amplitude and phase in the receive channels. There is no method to evaluate the measurement accuracy of the moving target velocity in the actual working condition of the radar system. To address this issue, we used an RTK UAV to simulate external reference targets, proposed a method to validate the weather radar radial velocity, and experimentally verified it using the X-band radar at the Changsha Meteorological Radar Calibration Center and the DJI M300 RTK UAV. Consistent with the application scenarios of the method, we introduced the evaluation parameter optimal absolute accuracy as a complementary parameter for assessing the validation of point targets. The experimental results indicate that the optimal absolute accuracy of the radar radial velocity validation is less than 0.05 m/s. Furthermore, the result is in close agreement with the velocity measurement results of the external instrument, confirming the effectiveness of this method in accurately reflecting the quality of the weather radar velocity. Compared with the conventional absolute validation accuracy, the proposed optimal absolute accuracy evaluation parameter in this method effectively reduces the validation errors caused by the instability of UAV attitudes. Moreover, the method can be effectively applied to the procedural handling of practical radial velocity validation.

The effect of the instability of the UAV on the radar velocity validation needs further quantitative study due to the limitations of the RTK system performance. (1) Influenced by the RTK system positioning frequency, data transmission, storage, etc., the UAV velocity update frequency is lower than the radar radial velocity acquisition frequency. Currently, only short-term average velocity can be compared as the instantaneous reference, and actual velocity references corresponding to each radar radial are not yet available. (2) The UAV RTK system may not operate effectively under some environmental conditions, such as harsh weather or complex terrain. This may limit the comprehensive assessment of radar validation. As RTK technology continues to improve and signal processing algorithms are optimized, future research will focus on aspects such as velocity comparisons based on high positioning frequency and the impact of sampling points and signal processing

algorithms on radial velocity validation. The aim is to improve the accuracy of the RTK UAV in the validation of the weather radar radial velocity.

**Author Contributions:** Conceptualization, Y.C. and L.L.; methodology, Y.C., L.L., and J.Z.; experiment, B.K. and M.Z.; process, L.L.; validation, F.Y.; formal analysis, Y.C.; resources, X.W; data curation, F.Y.; writing—original draft preparation, L.L.; writing—review and editing, Y.C.; visualization, Q.Y.; supervision, J.Z.; project administration, N.S.; funding acquisition, Z.B. All authors have read and agreed to the published version of the manuscript.

**Funding:** This research was funded by the National Key R&D Program of China (No. 2023YFC3007800; 2023YFC3007802) and the China Meteorological Administration Special Program for Innovation and Development (No. CXFZ2024Q008).

**Data Availability Statement:** Data is contained within the article.

**Acknowledgments:** The data acquisition of this field experiment was supported by the Changsha Meteorological Radar Calibration Center, for which we would like to express our sincere thanks!

**Conflicts of Interest:** The authors declare no conflicts of interest. The funders had no role in the design of the study; in the collection, analyses, or interpretation of data; in the writing of the manuscript; or in the decision to publish the results.

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
