# Peer review of "An RTK UAV-Based Method for Radial Velocity Validation of Weather Radar"

_remotesensing, doi:10.3390/rs16071153_

Round 1

Reviewer 1 Report

Comments and Suggestions for Authors

The authors should focus on the writing and formatting of the paper. The detailed review is attached.

Comments on the Quality of English Language

The paper is grammatically very poor.

Reviewer 2 Report

Comments and Suggestions for Authors

Comment 1 It is noted that your manuscript needs careful editing by someone with expertise in technical English editing paying particular attention to English grammar, spelling, and sentence structure so that the goals and results of the study are clear to the reader.

Comment 2 Considering the special environment of the RTK UAV, such as wind speed, altitude and other factors that may affect radial velocity measurements, a detailed discussion of these factors is required and the corresponding treatment methods are described.

Comment 3 The authors propose a validation index for the optimal absolute accuracy, but the original absolute accuracy index already meets the 1.0m/s requirement, so please explain in detail what advantages this index has compared to the absolute accuracy.

Comment 4 Please add a reference to the measurement indicator 1.0 m/s in table 8.

Comment 5 The methodology section of the paper describes in detail the research design, data collection and analysis methods. It is recommended that the authors further emphasize the innovation and applicability of the research methodology to enhance the academic value of the paper.

Comment 6 This work is summarized in the concluding section of the article, which invites further information on the limitations of the current methodology and directions that could be explored in the future, so that readers can build on this foundation for further research work.

Comments on the Quality of English Language

Moderate editing of English language required

Reviewer 3 Report

Comments and Suggestions for Authors

In this paper authors propose a novel method for weather radar radial velocity validation using RTK UAVs as external targets. It introduces the concept of Optimal Absolute Accuracy as an additional evaluation parameter and highlights its effectiveness in reducing systematic deviations.

Overall, this is a very well organized manuscript and authors did a thorough study and nice job explaining the methodology involved.

This manuscript is appropriate for the Remote Sensing journal and can be accepted with minor revisions.

1.      Combine the two sentences in lines 29-31. “Weather radar plays a crucial role in weather monitoring and disaster prevention, providing critical data for predicting and responding to extreme events.”

2.      Simplify the complex sentences in this manuscript.

For example, lines 48-50: rewrite "However, there is no method to evaluate the measurement accuracy of the moving target velocity in the actual working condition of the radar" as "Existing methods cannot assess the accuracy of moving target velocity measurements under real-world radar operating conditions."

There are many complex sentences in the manuscript that need to be revised.

3.      Provide a brief explanation or relevant literature for the rationale behind using Optimal Absolute Accuracy parameter.

4.      Consider removing sentence 545 as it is repetitive.

5.      Document specific limitations of the RTK system.

6.      Overall, the manuscript needs to be improved in terms of clarity (some sentences could be simplified and rephrased for better understanding).

Comments on the Quality of English Language

The manuscript needs to be improved in terms of clarity (some sentences could be simplified and rephrased for better understanding).

Round 2

Reviewer 2 Report

Comments and Suggestions for Authors

The authors have addressed all my concerns and I think this article can be accepted.